# Deep-Learning-Based Detection of Cranio-Spinal Differences between Skeletal Classification Using Cephalometric Radiography

**DOI:** 10.3390/diagnostics11040591

**Published:** 2021-03-25

**Authors:** Seung Hyun Jeong, Jong Pil Yun, Han-Gyeol Yeom, Hwi Kang Kim, Bong Chul Kim

**Affiliations:** 1Safety System Research Group, Korea Institute of Industrial Technology (KITECH), Gyeongsan 38408, Korea; shjeong@kitech.re.kr (S.H.J.); rebirth@kitech.re.kr (J.P.Y.); 2Department of Oral and Maxillofacial Radiology, Daejeon Dental Hospital, Wonkwang University College of Dentistry, Daejeon 35233, Korea; hangyeol1214@gmail.com; 3Department of Oral and Maxillofacial Surgery, Daejeon Dental Hospital, Wonkwang University College of Dentistry, Daejeon 35233, Korea; hwi1304@naver.com

**Keywords:** machine learning, artificial intelligence, malocclusion, diagnostic imaging

## Abstract

The aim of this study was to reveal cranio-spinal differences between skeletal classification using convolutional neural networks (CNNs). Transverse and longitudinal cephalometric images of 832 patients were used for training and testing of CNNs (365 males and 467 females). Labeling was performed such that the jawbone was sufficiently masked, while the parts other than the jawbone were minimally masked. DenseNet was used as the feature extractor. Five random sampling crossvalidations were performed for two datasets. The average and maximum accuracy of the five crossvalidations were 90.43% and 92.54% for test 1 (evaluation of the entire posterior–anterior (PA) and lateral cephalometric images) and 88.17% and 88.70% for test 2 (evaluation of the PA and lateral cephalometric images obscuring the mandible). In this study, we found that even when jawbones of class I (normal mandible), class II (retrognathism), and class III (prognathism) are masked, their identification is possible through deep learning applied only in the cranio-spinal area. This suggests that cranio-spinal differences between each class exist.

## 1. Introduction

Dentofacial dysmorphosis exhibits various aspects such as prognathism, retrognathism, maxillary hypoplasia, and asymmetry [1,2]. For their treatment, several techniques of orthognathic surgery or orthodontics are applied [2,3,4]. Meanwhile, the stomatognathic system is composed of static and dynamic structures, and its harmonious functioning is based on the balanced relationship between them [5]. In addition, hard and soft cephalic structures arise, grow, and organize in a mutual balance [6]. Cranio-facial skeletons constantly reflect these influences and their related functional conditions [1,6,7]. Therefore, the genesis of a malocclusion is usually linked to an impairment of some kind to eugnathic growth that involves to various extents the mandible, the maxilla, and the functional matrix (tongue and facial muscles) [5].

Until now, orthodontics and orthognathic surgery have mainly relied on linear and angular measurements for the diagnosis and the planning of the therapeutic procedures [1,3,7,8,9,10,11,12,13]. These measurements depend on the identification of several landmarks on cephalometric images, which are then applied to define the aforementioned measurements [1,3,7,8,9,10,11,12,13]. It is well recognized that the relation between these metrics varies with the type of bite and therefore is different in skeletal classes I, II, and III [1,7,13]. In addition, most of these landmarks on cephalometric images are concentrated in the maxilla and mandible [7]. However, the authors wondered if the difference between skeletal classes I, II, and III appears only in maxilla and mandible or if not, if is it also revealed in the cranio-spinal area excluding jaw. We also wanted to find a way to intuitively distinguish skeletal classes I, II, and III without linear and angular measurements.

Advances in convolutional neural networks (CNNs) continue [14,15,16,17]. They are being applied in a variety of dental and maxillofacial fields. For instance, they are used to assess soft-tissue profiling and extraction difficulty for mandibular third molars [18,19]. In addition, Xiao et al. proposed an end-to-end deep-learning framework to estimate patient-specific reference bony shape models for patients with orthognathic deformities [20]. Moreover, Sin et al. evaluated an automatic segmentation algorithm for pharyngeal airway in cone-beam-computed tomography images [21]. CNNs have proven their applicability to dental and maxillofacial fields through many other studies. However, to the best of our knowledge, CNNs have not been applied yet to clarify cranio-spinal differences between skeletal classification. Therefore, the aim of this study was to reveal cranio-spinal differences between skeletal classification using CNNs.

## 2. Materials and Methods

### 2.1. Datasets

In this study, transverse and longitudinal cephalometric images of 832 Korean patients who visited Daejeon Dental Hospital, Wonkwang University between January 2007 and December 2019 complaining about dentofacial dysmorphosis and/or a malocclusion were used for the training and testing of a deep-learning model (365 males and 467 females with a mean age of 18.37 ± 8.06 years). Patients with a congenital deformity, infection, trauma, or tumor history were excluded. The lateral and posterior–anterior (PA) cephalometric images were obtained using a Planmeca Promax (Planmeca OY, Helsinki, Finland), and the images were extracted in JPG format. The original images had a pixel resolution of 2045 × 1816 with a size of 0.132 mm/pixel.

All radiographic images were annotated by two orthodontists, two oral and maxillofacial surgeons, and one oral and maxillofacial radiologist. Point A–nasion–point B (ANB) and a Wits appraisal were used to diagnose the sagittal skeletal relationship. Jarabak’s ratio and Björk’s sum were used to determine the vertical skeletal relationship. With consensus of five specialists, patients’ skeletal type was determined: class I (*n* = 272, 111 males and 161 females with a mean age of 17.17 ± 8.28 years); class II (*n* = 294, 105 males and 189 females with a mean age of 19.47 ± 8.85 years); or class III (*n* = 266, 149 males and 117 females with a mean age of 18.36 ± 6.61 years).

The purpose of this study was to determine whether there is an additional structural difference that makes it possible to distinguish the skeletal class in the structures of the head and neck other than the jawbone. Thus, labeling was manually performed such that the jawbone was sufficiently masked while the parts other than the jawbone were minimally masked.

The PA cephalometric images were masked with three square markers: a lower large square containing maxilla and mandible (nasal floor and hard palate region~inferior border of mandible) plus right and left small squares containing the condylar process (Figure 1). 

The lateral cephalometric images were labeled with two square markers: a left long square containing the condylar process, coronoid process, mandibular ramus, and airway space and a right square containing the dentoalveolar region, maxilla, mandibular body, and lower facial soft tissue (Figure 2).

### 2.2. Preprocessing and Image Augmentation

Each patient’s PA and lateral cephalometric images were preprocessed for training. The acquired data were resized to have the same height and width for training because they were different for each patient. For image resizing, we used OpenCV’s API based on interpolation. Given that the skeleton is classified according to a geometric relationship, the height and width were resized at the same ratio. The height of the original cephalometric images was resized to 500, and the corresponding height ratio was applied to the width. After that, the cephalometric images were placed at the middle and zero-padding was performed to obtain 500 × 500 images. Note that the masking process mentioned in the dataset paragraph was applied after image resizing. In addition, data augmentation was performed for the preprocessed images to improve accuracy and prevent overfitting by using Pytorch’s color jitter and random horizontal flip. Finally, the data were normalized by using the following equation.
(1)$$p_{i,j}^{*}=\frac{\frac{p_{i,j}}{255}-mean}{std.}.$$
where *p_i,j_*^*^, *p_i,j_*, mean, and std. are normalized pixel value, original pixel value, mean, and standard deviation, respectively. The values of mean and std. are set to 0.5 and 0.5, respectively.

### 2.3. Architecture of the Deep CNN

The network structure that classifies into skeletal classes I, II, and III using PA and lateral cephalometric images is shown in Figure 3. The PA and lateral cephalometric images were fed to each feature extractor to obtain the feature map separately. Various backbone networks, such as VGG [22], ResNet [23], and DenseNet [24], can be used as feature extractors, and feature maps of different dimensions can be obtained according to each network’s structure. In this study, DenseNet, proposed in 2016, was used as a feature extractor. DenseNet is a network that extracts features by continuously connecting the feature map of the previous layers with the input of the next layer. Figure 4 shows a five-layer dense block with a growth rate *k* = 4. ResNet is a method consisting in adding feature maps, while DenseNet is a structure that concatenates feature maps. Through this structure, the vanishing gradient can be improved, and the feature propagation can be reinforced. The depth of the feature map extracted through DenseNet is determined according to the growth rate and the number of layers of each block, whereas the width and height are determined according to the number of downsamplings. In this study, because pretrained DenseNet121 was used, an input image of 500 × 500 × 3 is converted into a feature map of 15 × 15 × 1024 after passing through the feature extractor. Each feature map output from the PA and lateral cephalometric images is transformed into a vector through global average pooling and merged into one vector through concatenation. The final classification is performed through a dense layer. The proposed network was implemented using Pytorch 1.2.

### 2.4. Visualization Method

In this study, the feature map was displayed so that the part extracted as a feature of the PA and lateral cephalometric images could be confirmed. The class activation map proposed in 2015 was used as a display method [25]. The class activation map is calculated as the summation of each feature map multiplied by the corresponding weight value of the dense layer, as shown in Figure 5. Through this method, it is possible to check which part of the cephalometric image was activated for classification. The greater the activation, the redder it is; and the greater the inactivation, the bluer it is.

## 3. Results

The proposed CNNs were trained using the Adam optimizer [26]. The initial learning rate was set to 0.001. The learning rate decay was set to 0.95 and was applied every five epochs. To take into account the randomness of the deep-learning-network training algorithm, five random sampling crossvalidations were performed for two datasets. The average and maximum accuracy of the five crossvalidations were 90.43% and 92.54% for test 1 (evaluation of the entire posterior–anterior (PA) and lateral cephalometric images) and 88.17% and 88.70% for test 2 (evaluation of the PA and lateral cephalometric images obscuring the mandible). A box plot of the accuracy for each test is shown in Figure 6. Table 1 shows the confusion matrix for best accuracy result of each test.

## 4. Discussion 

The average and maximum accuracies of the five crossvalidations were 90.43% and 92.54% for test 1, and 88.17% and 88.70% for test 2. As expected, the predicted results were more accurate in test 1, in which all cephalometric images could be analyzed without masking the jawbone. However, the difference in accuracy between test 1 and test 2 was within 5%, which is not significant.

At the same time, with the class activation map, it is possible to know where the CNNs focused on the cephalometric images to provide a prediction (Figure 7). As might be expected, test 1 focused on the jawbone, especially on the state of the dentition. However, in test 2, the jawbone and dentition were obscured and could not be analyzed. Therefore, CNNs were forced to identify the remaining uncovered regions, that is, the cranio-spinal area. Figure 7 shows the wide area of the cranio-spinal area excluding the jawbone, which is hidden, marked in red. This reveals that the cranio-spinal area is discernibly different in skeletal classes I, II, and III.

## 5. Conclusions

In this study, we found that even when the jawbones of skeletal classes I, II, and III are masked, their identification is possible through deep learning applied only in the cranio-spinal area. This suggests that cranio-spinal differences exist between each class. Further research is required about where and how cranio-spinal differences emerge.

## Figures and Tables

**Figure 1 diagnostics-11-00591-f001:**
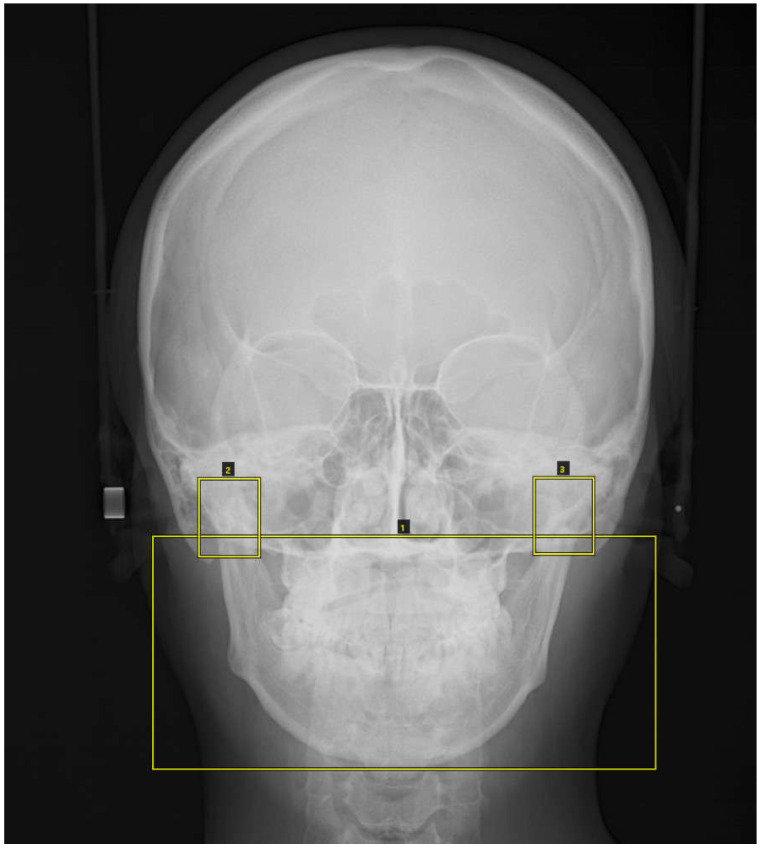
The posterior–anterior (PA) cephalometric images were masked with three square markers: a lower large square containing maxilla and mandible (nasal floor and hard palate region ~ inferior border of mandible) plus right and left small squares containing the condylar process.

**Figure 2 diagnostics-11-00591-f002:**
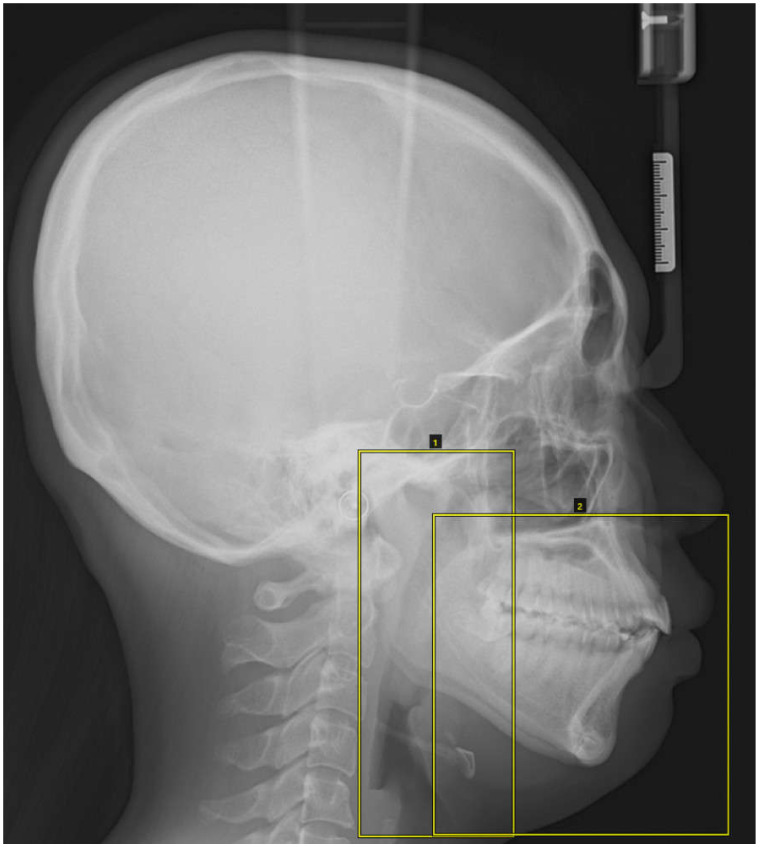
The lateral cephalometric images were masked with two square markers: a left long square containing the condylar process, coronoid process, mandibular ramus, and airway space and a right square containing the dentoalveolar region, maxilla, mandibular body, and lower facial soft tissue.

**Figure 3 diagnostics-11-00591-f003:**
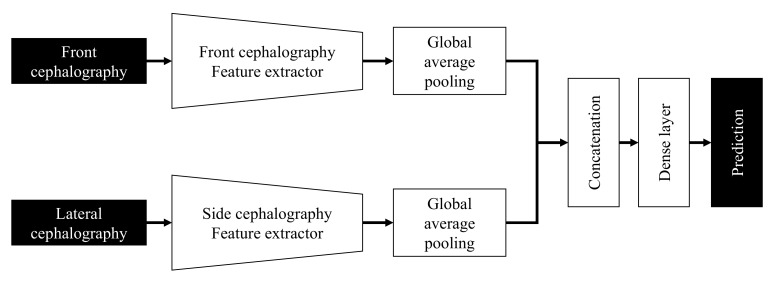
Multiside convolutional neural networks (CNNs) for classification using PA and lateral cephalometric images.

**Figure 4 diagnostics-11-00591-f004:**
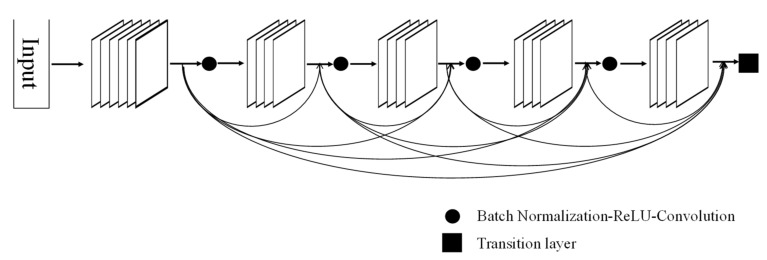
A five-layer dense block with a growth rate *k* = 4.

**Figure 5 diagnostics-11-00591-f005:**
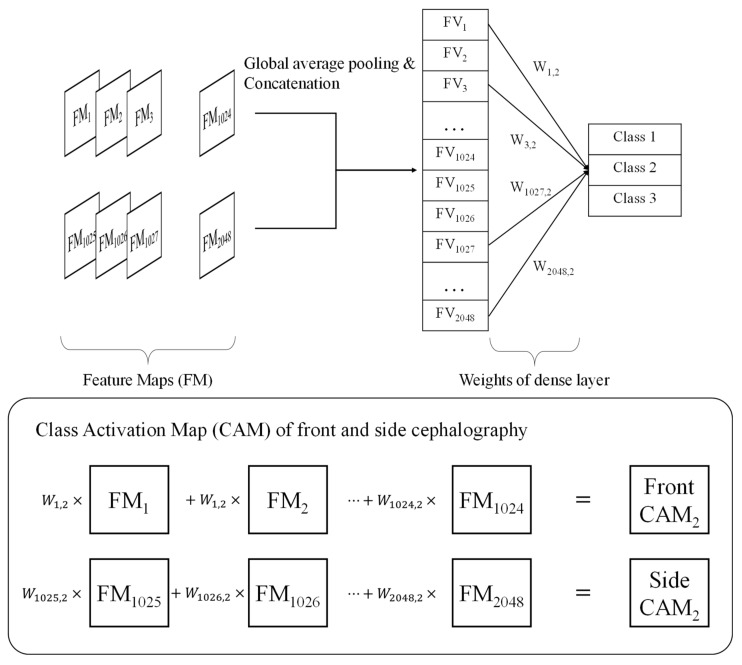
Class activation map (CAM) generation of PA and lateral cephalometric images.

**Figure 6 diagnostics-11-00591-f006:**
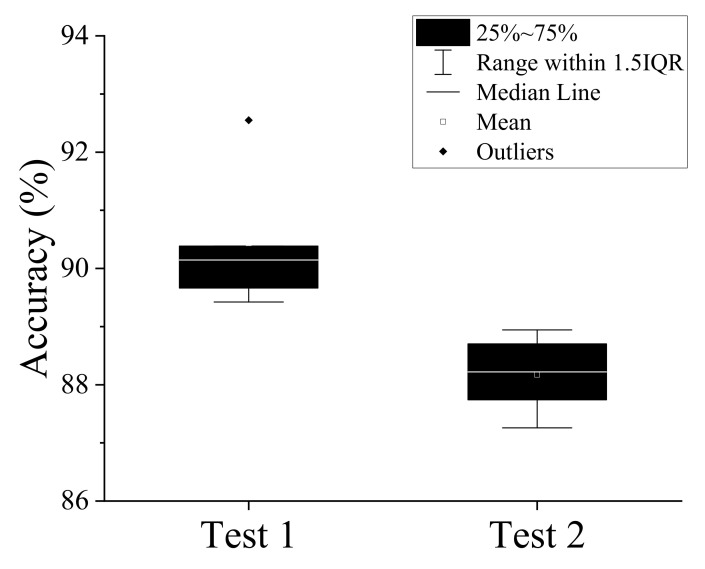
Test accuracies in five random sampling crossvalidations. Test 1: evaluation of the entire PA and lateral cephalometric images. Test 2: evaluation of the PA and lateral cephalometric images obscuring the mandible.

**Figure 7 diagnostics-11-00591-f007:**
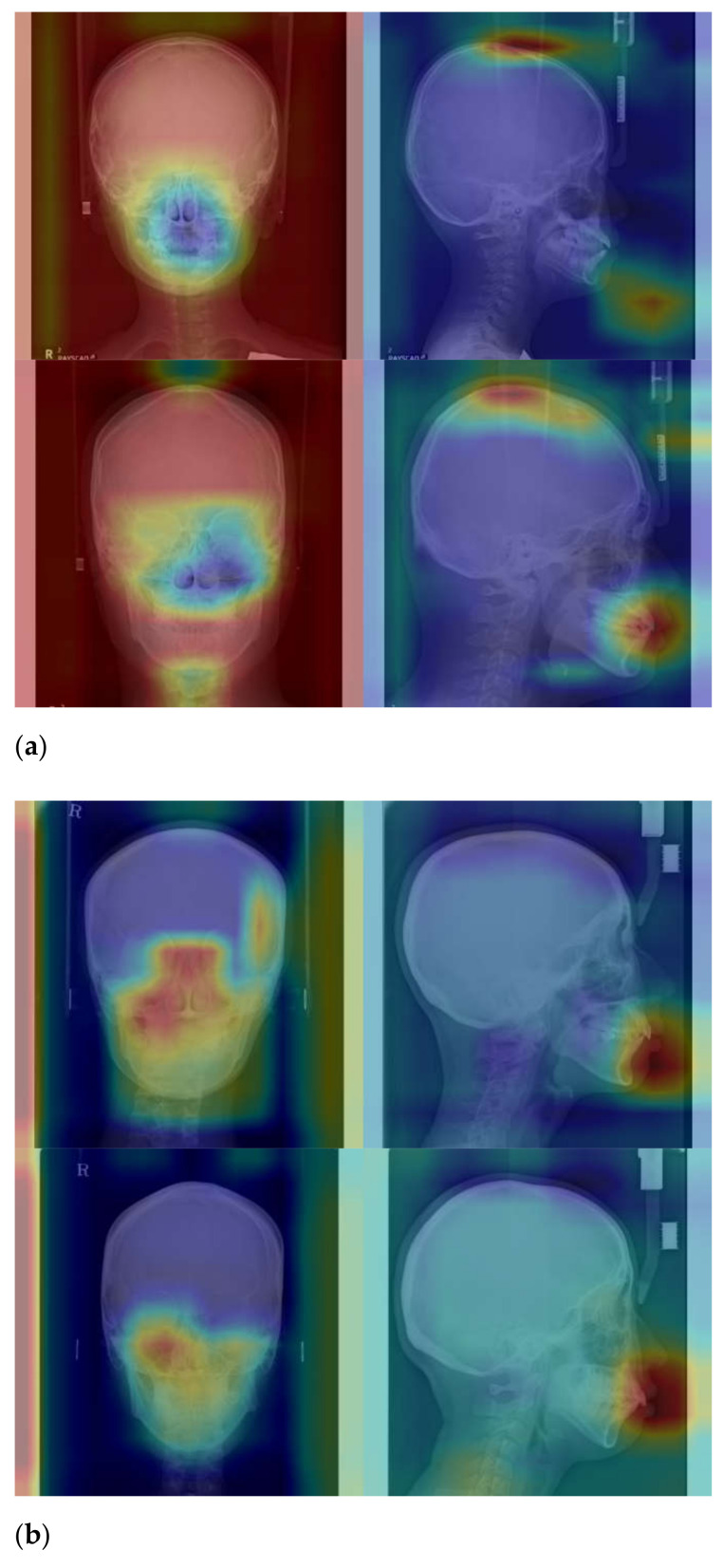
CAM of PA and lateral cephalometric images for (**a**) Class I of test 1, (**b**) Class II of test 1, (**c**) Class III of test 1, (**d**) Class I of test 2, (**e**) Class II of test 2, and (**f**) Class III of test 2. Class I: normal mandible; Class II: retrognathism; and Class III: prognathism. The greater the activation, the redder it is; the greater inactivation, the bluer it is.

**Table 1 diagnostics-11-00591-t001:** Confusion matrices of best accuracy results for (a) test 1 and (b) test 2.

(a)				
		**Predictions**		
		Class I	Class II	Class III
**Ground Truth**	Class I	125	9	7
	Class II	11	141	0
	Class III	3	1	119
(b)				
		Predictions		
		Class I	Class II	Class III
**Ground Truth**	Class I	118	12	8
	Class II	17	136	0
	Class III	8	2	115

Class I: normal mandible; Class II: retrognathism; and Class III: prognathism. Ground truth: actual group of patients classified according to their mandibular class. Prediction: mandibular class predicted by deep learning. Test 1: evaluation of the entire PA and lateral cephalometric images. Test 2: evaluation of the PA and lateral cephalometric images obscuring the mandible.

## Data Availability

The datasets generated during and/or analyzed during the current study are available from the corresponding author on reasonable request but is subject to the permission of the Institutional Review Boards of the participating institutions.

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
