# Peer review of "Deep-Learning-Based Detection of Cranio-Spinal Differences between Skeletal Classification Using Cephalometric Radiography"

_diagnostics, 2021, doi:10.3390/diagnostics11040591_

Round 1
Reviewer 1 Report
Dear Authors, thank you for submitting your paper.
The paper is interesting and can be considered for publication. It's well written and the topic is very original. Before publication, authors should add to their introduction and discussion the following articles:
https://doi.org/10.1016/j.sdentj.2020.09.001
https://doi.org/10.3390/nu12123688
https://doi.org/10.3390/ijerph17239104
-Data reported in the Methods section are appropriate and precisely described;.
-Results are reported clearly and adequately supported by Tables.
According to this Reviewer’s consideration, novelty and quality of the paper, publication of the present manuscript is recommended.
Author Response
Dear Authors, thank you for submitting your paper.
The paper is interesting and can be considered for publication. It's well written and the topic is very original. Before publication, authors should add to their introduction and discussion the following articles:
https://doi.org/10.1016/j.sdentj.2020.09.001
https://doi.org/10.3390/nu12123688
https://doi.org/10.3390/ijerph17239104
Ans)
We quoted the articles above in the introduction. (lines 30, 31, 33, 38)
Added sentences)
Dentofacial dysmorphosis exhibits various aspects such as prognathism, retrognathism, maxillary hypoplasia and asymmetry [1,2]. And for their treatment, several techniques of orthognathic surgery or orthodontics are applied [2-4]. Meanwhile, the stomatognathic system is composed of static and dynamic structures and its harmonious functioning is based on the balanced relationship between them [5]. In addition, hard and soft cephalic structures arise, grow, and organize in a mutual balance [6]. Cranio-facial skeletons constantly reflect these influences and their related functional conditions [1,6,7]. Therefore the genesis of a malocclusion is usually linked to an impairment of some kind to eugnathic growth that involves to various extents the mandible, the maxilla, and the functional matrix (tongue and facial muscles) [5].
-Data reported in the Methods section are appropriate and precisely described.
-Results are reported clearly and adequately supported by Tables.
According to this Reviewer’s consideration, novelty and quality of the paper, publication of the present manuscript is recommended.

Reviewer 2 Report
Overview:
The manuscript presents a deep learning procedure to classify the types of bite classes I, II, and III. The classification algorithm, which is a convolution neural network (CNN), uses as inputs PA and lateral cephalometric images. The authors found that even with large parts of the images masked it is possible to achieve excellent rates of classification, suggesting that there are skeletal differences in the skull that might be further explored. The manuscript is interesting and addresses an important aspect in orthodontics using a robust machine learning classifier. It is well written and easy to follow. Still, I have a few remarks that I detail next.
Comments/suggestions:
The introduction is rather short and the authors do not address the immense work that has been done in orthodontics regarding the determination of linear and angular measurements that are applied both on the diagnosis and on the planning of the therapeutic procedures. These measurements rely on the identification of several landmarks on cephalometric images, which are then applied to define the aforementioned measurements. It is well recognized that the relation between these metrics vary with the type of bite and therefore are different in classes I, II, and III. In my opinion, it is important to put the work in perspective considering what has been accomplished in the area. Only then, it is possible to assess how innovative is the work and how good are the results. Besides, the importance of masking parts of cephalometric images is only possible to understand completely if the referred landmarks are well identified. The introduction is also short concerning the application of machine learning techniques to dentistry and orthodontics. The reader should be informed at least about the success of these techniques and the importance of their application. My suggestion is to rewrite the introduction by revisiting some of the most important works in the area and giving enough details that allow a fair assessment of the results obtained.
The Materials and Methods section lacks some details that are important to understand the procedure and to evaluate the results. As classification is the main purpose of the CNN it is crucial to show that there is no bias in patients in each class, which could explain the results. Thus, age and sex should be described per class. Ethnicity is not referred in the manuscript but should also be described.
The authors refer that images were labeled with the consensus of 5 specialists (line 51), please indicate how the consensus was achieved.
Regarding the mask please indicate, which was the criterion to use those particular masks – for example, it seems that the nasion was not masked. Please, indicate also if the masks were placed manually or automatically. If they were manually placed, indicate how many people performed the task and how they proceeded.
Line 75: “The acquired data were resized”; please, indicate which algorithm was employed in the resizing.
In lines 80-81, the authors refer that “the cephalometric images were placed at the center”; please, define the center. Is it the middle of the image (point 250,250) or is it the center of mass?
Line 83, “data augmentation was performed for the pre-processed images”; please indicate the final number of images after data augmentation.
Line 85, “data were normalized so that the average value was 0.5”; please, indicate the equation of normalization.
Line 101, “an input image of 500 × 500 × 3”. I assume that as the authors used jpeg images they were codified as RGB. However, radiological images are intensity images having only one channel; therefore using three color channels is redundant. Please, clarify this point.
Table 1, presents the confusion matrices for tests 1 and 2, but as it was used cross-validation it is not clear if the matrices correspond to one of the test sets or some type of average of the 5 test sets of the cross-validation. Please, clarify this aspect by adding information in the table caption.
Finally, there is a typo in the title in the word “cephalometric”.
Author Response
Reviewer 2
Overview:
The manuscript presents a deep learning procedure to classify the types of bite classes I, II, and III. The classification algorithm, which is a convolution neural network (CNN), uses as inputs PA and lateral cephalometric images. The authors found that even with large parts of the images masked it is possible to achieve excellent rates of classification, suggesting that there are skeletal differences in the skull that might be further explored. The manuscript is interesting and addresses an important aspect in orthodontics using a robust machine learning classifier. It is well written and easy to follow. Still, I have a few remarks that I detail next.
Comments/suggestions:
The introduction is rather short and the authors do not address the immense work that has been done in orthodontics regarding the determination of linear and angular measurements that are applied both on the diagnosis and on the planning of the therapeutic procedures. These measurements rely on the identification of several landmarks on cephalometric images, which are then applied to define the aforementioned measurements. It is well recognized that the relation between these metrics vary with the type of bite and therefore are different in classes I, II, and III. In my opinion, it is important to put the work in perspective considering what has been accomplished in the area. Only then, it is possible to assess how innovative is the work and how good are the results. Besides, the importance of masking parts of cephalometric images is only possible to understand completely if the referred landmarks are well identified. The introduction is also short concerning the application of machine learning techniques to dentistry and orthodontics. The reader should be informed at least about the success of these techniques and the importance of their application. My suggestion is to rewrite the introduction by revisiting some of the most important works in the area and giving enough details that allow a fair assessment of the results obtained.
Ans) We fully agree with the reviewer's opinion. So, we thoroughly modified the introduction as follows: (lines 28-61)
Added sentences)
Dentofacial dysmorphosis exhibits various aspects such as prognathism, retrognathism, maxillary hypoplasia and asymmetry [1,2]. And for their treatment, several techniques of orthognathic surgery or orthodontics are applied [2-4]. Meanwhile, the stomatognathic system is composed of static and dynamic structures and its harmonious functioning is based on the balanced relationship between them [5]. In addition, hard and soft cephalic structures arise, grow, and organize in a mutual balance [6]. Cranio-facial skeletons constantly reflect these influences and their related functional conditions [1,6,7]. Therefore the genesis of a malocclusion is usually linked to an impairment of some kind to eugnathic growth that involves to various extents the mandible, the maxilla, and the functional matrix (tongue and facial muscles) [5].
Meanwhile, until now, orthodontics and orthognathic surgery mainly relies on linear and angular measurements for the diagnosis and the planning of the therapeutic procedures [1,3,7-13]. These measurements depend on the identification of several landmarks on cephalometric images, which are then applied to define the aforementioned measurements [1,3,7-13]. It is well recognized that the relation between these metrics vary with the type of bite and therefore are different in skeletal classes I, II, and III [1,7,13]. And most of these landmarks on cephalometric images are concentrated in maxilla and mandible [7]. However, the authors wondered whether the difference between skeletal classes I, II, and III appears only in maxilla and mandible, or if not, is it also revealed in the cranio-spinal area excluding jaw. We also wanted to find a way to intuitively distinguish skeletal classes I, II, and III without linear and angular measurements.
Advances in convolutional neural networks (CNNs) continue [14-17]. They are being applied in a variety of dental and maxillofacial fields. For instance, they are used to assess soft tissue profiling and extraction difficulty for mandibular third molars [18,19]. And Xiao et al. proposed an end-to-end deep learning framework to estimate patient-specific reference bony shape models for patients with orthognathic deformities [20]. Moreover, Sin et al. evaluated an automatic segmentation algorithm for pharyngeal airway in cone beam computed tomography images [21]. CNNs have proven their applicability to dental and maxillofacial fields through many other studies. However, to the best of our knowledge, CNNs have not been applied yet to clarify cranio-spinal differences between skeletal classification. Therefore, the aim of this study was to reveal cranio-spinal differences between skeletal classification using CNNs.
The Materials and Methods section lacks some details that are important to understand the procedure and to evaluate the results. As classification is the main purpose of the CNN it is crucial to show that there is no bias in patients in each class, which could explain the results. Thus, age and sex should be described per class. Ethnicity is not referred in the manuscript but should also be described.
Added sentences)
In this study, transverse and longitudinal cephalometric images of 832 Korean patients who visited Daejeon Dental Hospital, Wonkwang University, between January 2007 and December 2019 complaining about dentofacial dysmorphosis and/or a malocclusion, were used for the training and testing of a deep learning model (365 males and 467 fe-males with a mean age of 18.37±8.06 years). (line 64)
With consensus of 5 specialists, patients’ skeletal type was determined: class I (n=272, 111 males and 161 females with a mean age of 17.17±8.28 years), class II (n=294, 105 males and 189 females with a mean age of 19.47±8.85 years), or class III (n=266, 149 males and 117 females with a mean age of 18.36±6.61 years). (Materials and methods / 2.1. Datasets) (lines 77-80)
The authors refer that images were labeled with the consensus of 5 specialists (line 51), please indicate how the consensus was achieved.
Ans)
Point A–nasion–point B (ANB) and a Wits appraisal were used to diagnose the sagittal skeletal relationship. Jarabak’s ratio and Björk’s sum were used to determine the vertical skeletal relationship. One expert (Orthodontist) showed the analyzed images and propose the classification, and other experts discussed about the images and finish the labeling.
Regarding the mask please indicate, which was the criterion to use those particular masks – for example, it seems that the nasion was not masked. Please, indicate also if the masks were placed manually or automatically. If they were manually placed, indicate how many people performed the task and how they proceeded.
Ans)
- The masks were placed manually. The masked area was explained as following.
- The PA cephalometric images were masked with three square markers: a lower large square containing maxilla and mandible (nasal floor and hard palate region ~ inferior border of mandible) plus right and left small squares containing the condylar process (Figure 1).
- The lateral cephalometric images were labeled with two square markers: a left long square containing the condylar process, coronoid process, mandibular ramus, and airway space plus a right square containing the dentoalveolar region, maxilla, mandibular body, and lower facial soft tissue (Figure 2).
Added sentences)
Thus, labeling was manually performed such that the jawbone was sufficiently masked while the parts other than the jawbone were minimally masked. (Materials and methods, line 83)
Line 75: “The acquired data were resized”; please, indicate which algorithm was employed in the resizing.
Ans)
We used interpolation method provided by opencv api for resizing.
The sentence below is added in line 105.
For image resizing, we used opencv’s api based on interpolation.
In lines 80-81, the authors refer that “the cephalometric images were placed at the center”; please, define the center. Is it the middle of the image (point 250,250) or is it the center of mass?
Ans)
The first case – middle of the image is correct.
The sentence is revised as follows.
After that, the cephalometric images were placed at the middle and zero-padding was performed to obtain 500 × 500 images. (lines 109)
Line 83, “data augmentation was performed for the pre-processed images”; please indicate the final number of images after data augmentation.
Ans)
The data augmentation is performed by using dataloader api provided by pytorch. In detail, for each training iteration, the original batch images are sequentially transformed through colorjitter and random horizontal flip and fed to the neural network. Colorjitter, one of the image transform methods provided by pytorch, is a method that randomly adjusts brightness and contrast within a given value, the final number of images after data augmentation cannot be defined.
Line 85, “data were normalized so that the average value was 0.5”; please, indicate the equation of normalization.
Ans)
Sorry, the sentence was written incorrectly. We revise the sentence as follows.
(lines 114-118)
Finally, the data were normalized by using the following equation.
pi,j*=(pi,j/255 - mean)/std.,
where pi,j*, pi,j, mean and std. are normalized pixel value, original pixel value, mean, and standard deviation, respectively. The values of mean and std. are set to 0.5 and 0.5, respectively.
Line 101, “an input image of 500 × 500 × 3”. I assume that as the authors used jpeg images they were codified as RGB. However, radiological images are intensity images having only one channel; therefore using three color channels is redundant. Please, clarify this point.
Ans)
The radiological images have only one channel as you pointed out. However, in order to improve the performance of deep learning, it is effective to use pretrained weight values using a large dataset such as imagenet. Most of the pretrained weight values were obtained using 3 channel RGB images, so radiological images were expanded to 3 channels to use pretrained weights. To clarify the mentioned point, the sentence is revised as follows. (line 134)
In this study, because pre-trained DenseNet121 was used, an input image of 500 × 500 × 3 is converted into a feature map of 15 × 15 × 1024 after passing through the feature ex-tractor.
Table 1, presents the confusion matrices for tests 1 and 2, but as it was used cross-validation it is not clear if the matrices correspond to one of the test sets or some type of average of the 5 test sets of the cross-validation. Please, clarify this aspect by adding information in the table caption.
Ans)
The confusion matrices are best accuracy results for test 1 and test 2. By following the comment, I add the information in the table caption as follows.
Table 1. Confusion matrices of best accuracy results for (a) test 1 and (b) test 2.
(line 169)
Table 1 shows the confusion matrix for best accuracy result of each test. (line 163)
Finally, there is a typo in the title in the word “cephalometric”.
Ans)
We fixed it with “cephalometric”. (line 2)

Round 2
Reviewer 2 Report
The authors improved the manuscript and corrected the aspects pointed out. The answers provided in the cover letter fully explain the concerns I signaled in the first round of the revision. I have no further comments on the manuscript.